# A *Bidens pilosa* L. Non-Polar Extract Modulates the Polarization of Human Macrophages and Dendritic Cells into an Anti-Inflammatory Phenotype

**DOI:** 10.3390/molecules28207094

**Published:** 2023-10-14

**Authors:** Xandy Melissa Rodríguez Mesa, Leonardo Andres Contreras Bolaños, Geison Modesti Costa, Antonio Luis Mejia, Sandra Paola Santander González

**Affiliations:** 1Phytoimmunomodulation Research Group, Juan N. Corpas University Foundation, Bogotá 111161, Colombia; 2Phytochemistry Research Group (GIFUJ), Pontificia Universidad Javeriana, Bogotá 110231, Colombia

**Keywords:** *Bidens pilosa* L., immunomodulation, macrophages, dendritic cells, phytochemistry, traditional medicine, phytotherapy

## Abstract

Different communities around the world traditionally use *Bidens pilosa* L. for medicinal purposes, mainly for its anti-inflammatory, antinociceptive, and antioxidant properties; it is used as an ingredient in teas or herbal medicines for the treatment of pain, inflammation, and immunological disorders. Several studies have been conducted that prove the immunomodulatory properties of this plant; however, it is not known whether the immunomodulatory properties of *B. pilosa* are mediated by its ability to modulate antigen-presenting cells (APCs) such as macrophages (MØs) and dendritic cells (DCs) (through polarization or the maturation state, respectively). Different polar and non-polar extracts and fractions were prepared from the aerial part of *B. pilosa*. Their cytotoxic and immunomodulatory effects were first tested on human peripheral blood mononuclear cells (PBMCs) and phytohemagglutinin (PHA)-stimulated PBMCs, respectively, via an MTT assay. Then, the non-cytotoxic plant extracts and fractions that showed the highest immunomodulatory activity were selected to evaluate their effects on human MØ polarization and DC maturation (cell surface phenotype and cytokine secretion) through multiparametric flow cytometry. Finally, the chemical compounds of the *B. pilosa* extract that showed the most significant immunomodulatory effects on human APCs were identified using gas chromatography coupled with mass spectrometry. The petroleum ether extract and the ethyl acetate and hydroalcoholic fractions obtained from *B. pilosa* showed low cytotoxicity and modulated the PHA-stimulated proliferation of PBMCs. Furthermore, the *B. pilosa* petroleum ether extract induced M2 polarization or a hybrid M1/M2 phenotype in MØs and a semi-mature status in DCs, regardless of exposure to a maturation stimulus. The immunomodulatory activity of the non-polar (petroleum ether) extract of *B. pilosa* on human PBMC proliferation, M2 polarization of MØs, and semi-mature status in DCs might be attributed to the low–medium polarity components in the extract, such as phytosterol terpenes and fatty acid esters.

## 1. Introduction

Immunomodulation is a strategy that encompasses different therapeutic approaches to modify the immune response [1]. This therapeutic strategy has been used in conditions that require the activation of the immune response, such as infectious diseases, immunodeficiencies, and cancer, as well as in those that demand the control or reduction of the inflammatory response to treat allergies and autoimmune diseases or to prevent transplant rejection.

Different immunomodulatory therapies have focused on regulating the activity of cells of the innate (granulocytes and natural killer (NK) cells) or adaptive (T cells and B cells) immune system [2]. However, in the last 20 years, new therapies have begun to focus on the modulation of antigen-presenting cells (APCs) (macrophages (MØs) and dendritic cells (DCs)) [3,4]. Antigen-presenting cells modulate the activation and inactivation of innate and adaptive immune responses, as well as the balance between them [5,6]. It is well known that alterations in the phenotype and functionality of APCs favor the development of cancer and inflammatory and autoimmune diseases [6,7,8].

Macrophages are a functionally heterogeneous group of cells that can rapidly change their function in response to surrounding signals [6]. Cytokines and Toll-like receptor (TLR) agonists can induce macrophage activation and polarization (M1 or M2) of phenotypes, characterized by the expression of different gene and protein profiles [9]. M1 macrophages, essential for the development of effective antimicrobial and antitumor immune responses, display a proinflammatory phenotype distinguished by the secretion of tumor necrosis factor (TNF) alpha, nitric oxide (NO), inducible nitric oxide synthase (iNOS) interleukin (IL) 12, and IL-23, which favor Th1 and Th17 responses [6,10,11]. In contrast, M2 macrophages exhibit potent anti-inflammatory activity and antagonize the MØ M1 response [12,13]. M2 MØs play a crucial role in tissue homeostasis, wound healing, and fibrosis [14,15,16]. M2 macrophages also participate in the development of the Th2 responses against some fungi and helminths [6,17,18].

Dendritic cells are essential in triggering innate and adaptive immune responses and enforcing a protective tolerance to self-antigens [19]. The activation of immature DCs in tissues depends largely on the maturation status (immature, semi-mature, or mature) they acquire in response to a stimulus [20]. When DCs recognize an extracellular or intracellular stimulus that is insufficient to induce their activation, they remain immature. Immature DCs can efficiently capture antigens, although their processing and the resulting intracellular signaling do not increase the expression of molecules associated with naïve T cell activation for generating effector T cells. Therefore, immature DCs contribute to the induction of tolerance and control of immune responses [21]. On the other hand, immature DCs can mature upon recognizing pathogen- and damage-associated molecular patterns (PAMPs and DAMPs) through TLRs, collectins, or scavenger receptors. Mature DCs overexpress major histocompatibility complex (MHC) class I and II molecules, co-stimulatory molecules (CD80, CD83, CD86), release cytokines (IL-12 and IL-1ꞵ), and can induce clonal expansion of antigen-specific naïve T cells and their differentiation into effector T cells [19,22].

Several investigations have shown that some plant extracts and chemical compounds can activate or inhibit APC responses. For example, curcuminoids obtained from *Curcuma longa* L. or polysaccharides isolated from *Echinacea purpurea* (L.) Moench induce M2 and M1 polarization of MØs [23,24], respectively. Likewise, terpenes and flavonoids decrease DC activation, while polysaccharides such as acemannan obtained from *Aloe* vera (L.) Burm.f., angelane from *Angelica gigas* Nakai, and galactomannan from *Caesalpinia spinosa* induce DC maturation [25,26,27].

*Bidens pilosa* is a plant traditionally used in tropical and subtropical areas around the world to reduce inflammation and pain. These uses have been recorded in America, Africa, Asia, and Oceania [28]. Currently, several scientific works have demonstrated and confirmed that different chemical compounds of this plant modulate the synthesis of inflammatory mediators (IL-1β, IL-6, TNF-α) and reactive oxygen species (ROS) in vitro [29,30,31]. In addition, in vivo studies in animal models have shown that the anti-inflammatory and antinociceptive properties that *B. pilosa* are associated with include polyacetylenes (PCs) and phenolic compounds [32,33,34].

Although different extracts and chemical compounds of *B. pilosa* have been shown to modulate the innate immune response, their ability to regulate the activation or inhibition of APCs is unknown. Therefore, this study aimed to evaluate the effect of different extracts and fractions of *B. pilosa* on MØ polarization and DC maturation.

## 2. Results and Discussion

### 2.1. Bidens pilosa Extracts and Fractions Showed No Cytotoxic Activity towards Human Peripheral Blood Mononuclear Cells and Modulated Their Proliferation and Cytokine Secretion

None of the tested extracts and fractions of *B. pilosa* were cytotoxic to human PBMCs in either of the three evaluations, and their IC_50_ values were above the theoretical figure considered cytotoxic (IC_50_ < 100 µg/mL) (Figure 1A). The activity of all *B. pilosa* extracts and fractions was tested on PHA-stimulated human PBMC cultures for 12, 24, and 48 h; however, only the treatments that had the most significant effects are shown at 48 h.

In PBMCs stimulated and treated at the same time, the PI decreased by 23 to 65% in cultures exposed to the petroleum ether extract (2) (1.56 to 200 µg/mL), by 10 to 45% in those treated with the ethyl acetate fraction (3.3) (1.56 to 200 µg/mL), and by 2 to 14% in cultures exposed to the hydroalcoholic fraction (25 to 200 µg/mL), respectively (Figure 1B). Additionally, cells exposed with this hydroalcoholic fraction at lower concentrations (1.56 to 12.5 µg/mL) showed an increase in the IP by 9% to 3%, respectively (Figure 1B).

Furthermore, the production of proinflammatory cytokines (IL-1β, IL-6, IL-8, and TNF) in PHA-stimulated PBMC cultures was modulated by some of the treatments derived from *B. pilosa*. In particular, IL-6 was significantly decreased in the presence of the petroleum ether extract (2) and the hydroalcoholic fraction (3.4) (* *p* < 0.05); the latter also induced a decreased production of IL-1β (* *p* < 0.05). IL-10 and IL-12p70 were not detected in the supernatants of any culture or control.

The addition of the petroleum ether extract (2) (1.56 to 200 µg/mL) to already proliferating PBMCs decreased the PI by 4% to 14% and the ethyl acetate (3.3) and hydroalcoholic (3.4) fractions by 9% to 19% and 8% to 18%, respectively (Figure 1C). These treatments significantly decreased TNF levels (** *p* < 0.01 and *** *p* < 0.001, respectively) but did not alter the concentration of IL-1β, IL-6, or IL-8 in culture supernatants.

*B. pilosa* has previously been reported to have low cytotoxic activity, even in tumor cell lines. Kviecinski et al. demonstrated that high concentrations of *B. pilosa* extracts of low and intermediate polarity—obtained through a supercritical fluid process—decreased the viability of human breast cancer MCF7 cells. Furthermore, they calculated the IC_50_ values of such extracts to be between 437 μg/mL and 291 μg/mL after 24 h and 48 h incubation [35]. Similarly, an aqueous (polar) extract of *B. pilosa* var. minor Sheriff was found to be slightly cytotoxic to human leukemic cell lines (L1210, U937, K562, Raji, and P3HR1), showing IC_50_ values between 145 μg/mL and 586 μg/mL [36].

Some authors have shown that a crude methanolic extract of *B. pilosa* containing medium- and high-polarity compounds inhibited the proliferation of human PBMC stimulated with PHA. In contrast, other authors demonstrated that the aqueous infusion of *B. pilosa* increases cytokine production (TNF and IL-1β) by human whole blood, both in the presence or not in the presence of lipopolysaccharide (LPS) [37].

This last result could be associated with the activity of PCs which, as observed in models of *Listeria monocytogenes*, *Candida parapsilosis*, and *Eimeria tenella* infection, enhance the phagocytic activity of macrophages and increase the production of IFN-γ by T cells [34,38,39,40]. On the other hand, in in vivo models, the anti-inflammatory potential of *B. pilosa* has been determined. In a colitis animal model, aqueous extract decreased the leukocytic infiltration in the intestine and the production of inflammatory cytokines such as TNF-α [33,41].

These assays reporting anti-inflammatory or proinflammatory responses may be associated with the activity of different secondary or primary metabolites present in the extracts, which may generate one response or the other. In our assays, extract 2, which is enriched with secondary metabolites, exhibits anti-inflammatory activity that is also reported in the literature (Table 1 and Table 2).

Furthermore, the pharmaceutical product FITOPROT (*B. pilosa* extract combined with curcuminoids) can reduce the synthesis of proinflammatory cytokines (IL-1β, IL-6, IL-8, and TNF-α) by human HaCaT keratinocytes exposed to 5-fluorouracil (5FU) [29].

### 2.2. The Non-Polar Extract and the More Polar Fractions Obtained from B. pilosa Promoted an Anti-Inflammatory Profile of Human Macrophages and Dendritic Cells

Petroleum ether extract (2), ethyl acetate (3.3), and hydroalcoholic (3.4) fractions obtained from *B. pilosa* were selected for further studies on their activity on human MØs and DCs.

Notably, the *B. pilosa* petroleum ether extract (2) polarized M1-preconditioned MØs into the M2 phenotype (CD80^Low^/CD86^Low^/CD163^High^/CD206^Med^/CD209^High^/iNOS^Low^) and increased the IL-10 levels in culture supernatants (Figure 2).

Moreover, the *B. pilosa* ethyl acetate (3.3) and hydroalcoholic (3.4) fractions polarized M1-preconditioned MØ towards an “incomplete” M2 phenotype (CD80^Low^/CD86^Low^/CD163^Med/Low^/CD206^Med^/CD209^Med/Low^/iNOS^Low^) and induced a simultaneous increase in IL-10 production. Importantly, none of the extracts or fractions of *B. pilosa* L. analyzed could polarize M1-preconditioned MØ into the proinflammatory phenotype (CD80^High^/CD86^High^/CD163^Low^/CD206^Low^/CD209^Low^ /iNOS^High^) or increase the levels of IL-1β, IL-12p70, and TNF in culture supernatants (Figure 2).

Furthermore, after being added to the M2-preconditioned MØ cultures, the petroleum ether extract (2) of *B. pilosa* induced a significant increase in IL-8 and IL-10 (Figure 3), cytokines associated with an anti-inflammatory profile M2 [42,43,44,45]. This finding agrees with the results described above, according to which this non-polar extract of *B. pilosa* showed an M2-polarizing effect on M1-preconditioned MØs (Figure 3A,C).

The ethyl acetate (3.3) and hydroalcoholic (3.4) fractions of *B. pilosa* also polarized the M2-preconditioned MØs to an “incomplete” M2 phenotype (CD80^Low^/CD86^Low^/CD163^Mid/Low^/CD206^Mid^/CD209^Mid/Low^/iNOS^Low^). In addition, fraction 3.3 induced the production of IL-6 and IL-10, whereas fraction 3.4 favored the synthesis of IL-6, IL-8, and IL-10. It should be noted that IL-6 has also been associated with the development of anti-inflammatory MØs in both in vitro and in vivo models of neuroinflammation [46,47] (Figure 3). These results indicate that the petroleum ether extract (2) of *B. pilosa* enhances the complete switching of MØs into the M2 anti-inflammatory phenotype, regardless of the growth stimuli present during their differentiation.

Moreover, immature DCs exposed to the petroleum ether extract (2) of *B. pilosa* expressed a phenotypic profile identified as CD206^High^/CD209^High^/CD83^High^/CD86^High^/HLA-DR^High^. A phenotypic change (CD206^High^/CD209^High^/CD83^Med^/CD86^Med^/HLA-DR^Med/Low^) was also observed in cultures exposed to the plant-derived fractions 3.3 and 3.4.

None of the *B. pilosa* extracts or fractions tested were found to induce the synthesis of proinflammatory cytokines (IL-1β, IL-12p70, and TNF) when added to immature DC cultures (Figure 4D).

Altogether, these data indicated that the petroleum ether extract of *B. pilosa* is the only treatment that favored the progression of immature DCs into a “semi-mature” status, characterized by the increased expression of major histocompatibility complex class II (MHC-II) and co-stimulatory (CD83 and CD86) molecules, and no increase in the synthesis of the proinflammatory cytokines that were evaluated [48] (Figure 4).

It is well known that immature and semi-mature DCs induce T cell anergy and, thus, tolerance to peripheral self-antigens. These observations have opened new perspectives for developing therapies against allergies and autoimmune diseases and allergies [48,49].

Therefore, the finding that the petroleum ether extract of *B. pilosa* induced immature DCs to progress to a semi-mature status makes this phytochemical a promising therapeutic compound that should be tested in contexts of chronic inflammation where regulation of the immune response is highly required.

Given the anti-inflammatory properties of the petroleum ether extract (2) of *B. pilosa* on the polarization into M2 MØs, additional assays were performed to confirm its activity in the polarization of M1-inflammatory MØs (stimulated with IFN-γ) towards M2. The results showed that this extract favored the co-expression of CD209 and CD206 associated with an M2 profile but did not significantly decrease the expression of HLA-DR, CD86, or iNOS, with a production of IL12p70, TNF, IL-6, and IL-8 (Figure 5A). Therefore, macrophages treated with petroleum ether show a hybrid M1/M2 phenotype (CD206^High^/CD209^High^/CD86^Med^/iNOS^Med/^). The functionality of the M1/M2 phenotype of macrophages has not yet been well described; however, some authors describe that this profile would be associated with the regulation of the assembly and structure of the extracellular matrix, which is essential during the repair process and homeostasis of the tissues [50].

In the same way, and taking into account that the petroleum ether extract (2) on DCs induces the semi-maturation of these cells, (Figure 4) it was established if this extract can or cannot favor their maturation in co-treatment with a maturation stimulus such as LPS. The results obtained showed that the petroleum ether extract of *B. pilosa* does, indeed, change the mature phenotype induced for LPS (CD206^Med^/CD209^Med^/CD83^High^/CD86^High^/HLA-DR^High^) toward a phenotype status CD206^High^/CD209^High^/CD83^High^/CD86^High^/HLA-DR^Med^; additionally, treatment only modulated a reduced TNF production, without affecting production of other cytokines when compared with the vehicle control. Since this profile is similar to a semi-mature profile, the extract seems to be inhibiting the complete maturation of dendritic cells.

The observed results show that *Bidens* apolar extract has anti-inflammatory potential that could be used in future therapeutic approaches for the treatment of chronic inflammatory conditions, such as autoimmune diseases, in which APCs play an important role in the chronicity and maintenance of these conditions [51,52,53] due to the overproduction of proinflammatory cytokines (IL-1β, IL-6, and TNF-α) and overexpression of some transcription factors, such as the nuclear factor kappa B (NF-κB) and the signal transducer and activator of transcription 3 (STAT3) [54].

The development and chronicity of several autoimmune diseases (systemic lupus erythematosus (SEL), primary Sjögren’s syndrome, thyroid autoimmunity, type I diabetes (T1D), and rheumatoid arthritis (RA)) are associated with a constant synthesis of IL-1β [55,56]. This cytokine affects cell survival by decreasing the DNA content and reducing protein synthesis, metabolism, energy production, and the cell cycle, thus favoring the development of autoimmune disorders [57]. For example, elevated serum levels of IL-1β along with those of TNF-α have been associated with increased pancreatic β-cell destruction in patients with T1D, at all stages of the disease [58,59,60,61,62].

High levels of IL-1β are also observed in the serum and cerebrospinal fluid (CSF) of patients with multiple sclerosis. IL-1β alters the excitation–inhibition balance of neurons, and its elevated levels in CSF are associated with brain lesions [63,64]. These observations corroborate the relationship between chronic IL-1β production and disease severity. Furthermore, in patients with RA, a synergistic interaction between TNF-α, IL-1β, and IL-6 is related to disease progression, as it promotes chronic joint inflammation, constant activation of proinflammatory MØs, and erosive changes in cartilages and bones [65,66,67]. Similarly, TNF-α, IL-1β, and IL-6 secreted by MØs also play an essential role in the pathophysiology of autoimmune inflammatory bowel diseases such as colitis. These cytokines accumulate in intestinal tissue and induce extracellular matrix degradation, epithelial damage, endothelial activation, and blood vessel disruption [68,69,70].

Altogether, the above evidence supports the critical role of proinflammatory cytokines in the progression of autoimmune diseases. Some clinical trials with antibodies antagonizing the biological activity of cytokine, such as tocilizumab (anti-IL-6R mAb), have yielded effective therapeutic results for inflammatory autoimmune disorders such as RA and systemic and polyarticular juvenile idiopathic arthritis [71].

Therefore, the search for new natural immunomodulators capable of controlling or decreasing the proliferation and differentiation of APC cells and, thus, the synthesis of cytokines responsible for maintaining chronic inflammation has become an important field of research for managing autoimmune diseases.

This is the first study that shows the anti-inflammatory properties of *B. pilosa* extracts on MØ polarization and the DCs’ maturation status. Currently, our research group is evaluating the cell signaling pathways involved in *Bidens* activity on these cells, which will complement the characterization of this activity and allow us to progress in the pre-clinical models associated with the control of inflammation in autoimmune diseases.

### 2.3. The Medium–Low Polarity Compounds of the Petroleum Ether Extract of B. pilosa Would Be Associated with the Induction of Anti-Inflammatory Human Macrophages and Dendritic Cells

The petroleum ether extract of *B. pilosa* showed the highest immunomodulatory activity among all the extracts and fractions of *B. pilosa* tested here. Therefore, this extract was chemically analyzed via GC-MS and was shown to contain 17 compounds. These were mainly terpenes (48%), phytosterols (27%), fatty acids (14%), and aliphatic compounds (9%).

Some of the molecules identified in the petroleum ether extract of *B. pilosa,* such as germacrene D, (+)-Spathulenol, α-Humulene epoxide II, Ethyl palmitate, Methyl linolenate, Ethyl linoleate, Ethyl linolenate, Stigmasterol, Beta-Sitosterol, and Friedelan-3-one, have been previously reported by other authors (Table 1) [28,72].

**Table 1 molecules-28-07094-t001:** Compounds of the petroleum ether extract of *B. pilosa* previously reported to have immunomodulatory activity. Structures taken from: pubchem.ncbi.nim.nih.gov (accessed on 7 February 2023).

RT(min)	Abundance(%)	Similarity(%)	Compound	Structure	Biological Activity	Reference
8.677	1.16	92	Germacrene D	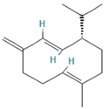	Anti-inflammatory. Inhibits calcium mobilization, chemotaxis, and ROS production in neutrophils.	[73]
9.993	2.60	89	(+)-Spathulenol	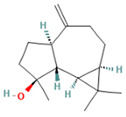	Antinociceptive, antiproliferative activity against lymphocytes by inducing apoptosis.	[74]
10.063	4.29	88	Caryophyllene epoxide	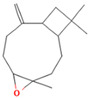	Anti-tumor, anti-inflammatory, analgesic.	[75,76]
10.273	1.74	91	α-Humulene epoxide II	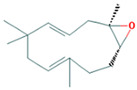	Antioxidant, anti-inflammatory.	[77,78]
13.337	3.25	90	Ethyl palmitate	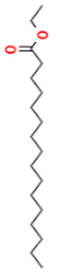	Anti-inflammatory, anti-arthritic, and immunomodulatory. In vivo animal model: decreases PGE2 levels, plasma levels of TNF-α, and IL-6. Reduces NF-κB expression in liver and lung tissues and neutrophil tissue infiltration.	[79,80]
14.797	1.18	85	Methyl linolenate	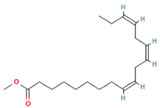	Antioxidant and anti-inflammatory, decreases NO, iNOS, COX-2, and IL-1β synthesis in LPS-stimulated MØs.	[81,82]
15.577	3.60	83	Ethyl linoleate	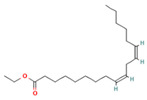	Anti-inflammatory. Down-regulates iNOS and COX-2 expression, reduces NO and PGE2 production in LPS-activated RAW 264.7 cells.	[83]
15.673	5.34	87	Ethyl linolenate	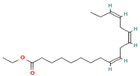	-	-
27.620	16.06	85	Stigmasterol	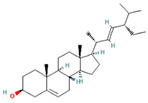	Antimicrobial, antioxidant.In vitro anti-inflammatory activity against newborn mouse chondrocytes and primary cultures of IL-1β-stimulated patient-derived chondrocytes. Inhibits the NF-kB pathway and several proinflammatory mediators and mediators of matrix degradation. Anti-inflammatory activity for microglia.	[84,85]
28.447	10.32	86	Beta-Sitosterol	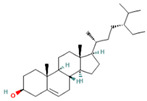	Anti-inflammatory.Alleviates inflammatory response by inhibiting activation of ERK/p38 and NF-κB pathways in LPS-exposed BV2 cells. Antiangiogenic activity in in vitro and in vivo models of rheumatoid arthritis.	[86,87]
32.44	29.10	85	Friedelan-3-one	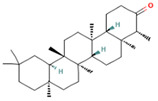	Antioxidant, anti-inflammatory, and immunomodulatory activities.In vivo model: decreases DSS-induced colitis and regulates autophagy. It is associated with the inhibition of ROS production (intracellular/extracellular), TNF-α, IL-1β, and IL-6.It also inhibits cell proliferation.	[88,89]

Abbreviations: COX-2: Cyclooxygenase-2; DSS: Dextran sulfate sodium; IL: Interleukin; HO-1: Heme oxygenase-1; iNOS: Inducible nitric oxide synthase; LPS: Lipopolysaccharide; NF-κB: nuclear factor kappa B; NO: Nitric oxide; PGE2: Prostaglandin E2; ROS: Reactive oxygen species; RT: Retention time; TNF-α: Tumor necrosis factor alpha.

Nevertheless, novel compounds were also identified in the petroleum ether extract of *B. pilosa*, namely dihydroactinidiolide, ent-germacra-4(15),5,10(14)-trien-1.beta-ol, neophytadiene, 1,1,6-trimethyl-3-methylene-2-(3,6,9,13-tetramethyl-6-ethene-10,14-dimethylene-pentadec-4-enyl) cyclohexane, tetrapentacontane, and methylcomate C (Table 2).

**Table 2 molecules-28-07094-t002:** Novel compounds in the petroleum ether extract of *B. pilosa* with immunomodulatory activity. Structures taken from: pubchem.ncbi.nim.nih.gov (accessed on 7 February 2023).

RT(min)	Abundance(%)	Similarity(%)	Compound	Structure	Biological Activity	Reference
9.610	0.64	93	Dihydroactinidiolide	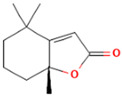	Antioxidant activity.	[90]
10.847	2.23	82	ent-Germacra-4(15),5,10(14)-trien-1.beta.-ol	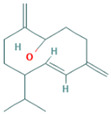	-	-
11.833	1.29	88	Neophytadiene	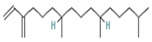	Anti-inflammatory activity. Inhibits the production of NO and the inflammatory cytokines TNF-α, IL-6, and IL-10 in in vitro (RAW 264.7 cells) and in vivo (Sprague Dawley rats) models.	[91]
22.897	1.95	85	1,1,6-trimethyl-3-methylene-2-(3,6,9,13-tetramethyl-6-ethenye-10,14-dimethylene-pentadec-4-enyl) cyclohexane	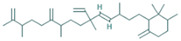	Antiproliferative activity against tumor cells.	[92]
23.457	8.83	83	Tetrapentacontane	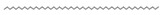	Antioxidant and anti-inflammatory activities.	[93]
29.123	5.15	89	Methyl commate C	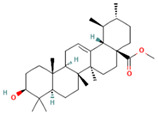	Antioxidant, anti-inflammatory, antidiabetic, and antihyperlipidemic activities.	[94,95]

Abbreviations: IL: Interleukin; NO: Nitric oxide; ROS: Reactive oxygen species; RT: Retention time; TNF-α: Tumor necrosis factor alpha.

Some of the terpenes of *B. pilosa* identified in this study have anti-inflammatory activity. Friedelan-3-one, a pentacyclic triterpene, abounds in the petroleum ether extract of *B. pilosa* and has also been isolated from other plants, such as *Azima tetracantha* Lam. It exhibits anti-inflammatory, analgesic, and antipyretic effects in animal models of inflammation [96]. For example, it counteracts dextran sodium sulfate (DSS)-induced murine colitis by reducing IL-1β and IL-6 synthesis and potentiating IL-10 synthesis. Furthermore, according to network pharmacology integrated with bioinformatics, friedelan-3-one had several potential targets, such as the C-C chemokine receptor type 2 (CCR2) [89]. This receptor is essential for promoting macrophage M1/M2 polarization and inducing DC maturation, as it up-regulates the expression of costimulatory molecules [97,98]. Thus, friedelan-3-one could exert its immunomodulatory activity on MØs and DCs by binding to the CCR2 receptor.

Essential oils enriched in terpenes derived from *Cannabis sativa* L. (β-pinene, myrcene, cis-α-bergamotene, α-humulene, trans-caryophyllene) show anti-inflammatory activity on the murine macrophage RAW 264.7 cell line by inhibiting nitric oxide radical production [99]. Similarly, another terpene, neophytadiene, isolated from the marine algae *Turbinaria ornata* (50 and 100 μM/mL) inhibits NO production and induces down-regulation of iNOS and NF-κB in murine macrophage RAW 264.7 stimulated with LPS and also in an in vivo model of inflammation induced by LPS in Sprague Dawley rats [91].

Beta-caryophyllene modulates the cannabinoid CB2 receptor (CB2R) involved in M2 polarization of MØs and DC maturation [52,100,101]. However, the activity of oxidized caryophyllene on these APCs remains unclear. Although oxidized caryophyllene has no affinity for CB2R, it has shown anti-inflammatory and analgesic properties [75,76] and the ability to modulate the STAT3-mediated signaling pathway in tumor cells. Considering that this pathway is also important for MØ polarization and DC maturation [102,103,104], it is possible to hypothesize that oxidized caryophyllene could act on MØs and DCs by modulating STAT3-mediated pathways; however, studies are needed to confirm this mechanism of action.

Stigmasterol was another abundant compound identified in the petroleum ether extract of *B. pilosa*. It is an anti-inflammatory phytosterol that decreases neuroinflammation in in vivo and in vitro models of Alzheimer’s disease. Specifically, it attenuates the inflammatory response of microglia via NF-κB signaling and nucleotide-binding domains, leucine-rich–containing family pyrin domain–containing-3 (NLRP3) via adenine monophosphate activated protein kinase (AMPK) activation [105]. Furthermore, stigmasterol inhibits phagocytosis, NO, and TNF-α production, Cyclooxygenase (COX-2) expression, iNOS, and phosphorylated extracellular signal-regulated protein kinase (p-ERK) in LPS-activated RAW264.7 cells [106]. These findings indicate that stigmasterol could strongly inhibit M1 polarization of MØs. However, M2 polarization was not assessed in any of these studies. No data on the activity of stigmasterol on human DCs were found in the literature reviewed.

Beta-sitosterol, a sterol commonly found in plants, exhibits anti-inflammatory activity. It alleviates the inflammatory response by inhibiting the activation of ERK/p38- and NF-κB-mediated pathways in LPS-exposed microglia cells [86]. Beta-sitosterol also attenuates inflammation in an animal model of RA by modulating M2 polarization of MØs [107]. The mechanism of action responsible for this immunomodulatory activity could be associated with the affinity of beta-sitosterol to the glucocorticoid receptor (GR). The structural similarity of sitosterol to steroids and corticosteroids would explain its ability to reduce airway inflammation in in vivo models of asthma [108]. Although beta-sitosterol is not an exclusive compound of medicinal plants with immunomodulatory activity, it could act in synergy with other anti-inflammatory molecules identified in the petroleum ether extract of *B. pilosa* and potentiate their effects.

Macrophages express the GR; glucocorticoids such as dexamethasone act on LPS-stimulated MØs by inhibiting the p38 MAPK pathway mainly activated in M1 MØs [109,110]. A GR-mediated anti-inflammatory effect has also been evidenced in DCs that acquire tolerogenic properties when exposed to glucocorticoids [83].

Fatty acids are another group of compounds in the petroleum ether extract of *B. pilosa* with potential immunomodulatory activity on APCs. The literature on metabolic networks and signaling pathways involved in MØ polarization indicates that both the availability and type of substrates in the surrounding media are critical for the functionality of these cells [111]. Saturated fatty acids are known to be proinflammatory and induce M1 MØs through TLR2 and TLR4 activation. In contrast, unsaturated fatty acids inhibit these TLR-dependent signaling pathways and act as ligands of PPAR-γ, thus favoring M2 polarization of MØs [112,113,114,115]. Ethyl linoleate was a fatty acid identified in the petroleum ether extract of *B. pilosa*. It is also isolated from *Allium sativum* and down-regulates iNOS and COX-2 expression in LPS-activated RAW 264.7 cells [83]. Another fatty acid, ethyl palmitate, is reported to ameliorate carrageenan-induced rat paw edema, and reduce PGE2 levels, plasma TNF-α and IL-6 concentrations, and NF-κB expression in liver and lung cells, as well as neutrophil tissue infiltration in a rat model of LPS-induced endotoxemia [80].

Given the modulatory activity of the phytochemicals described above, it is possible to postulate that the terpenes, phytosterols, and fatty acids identified in the petroleum ether extract of *B. pilosa* have anti-inflammatory effects that could also modulate human MØ polarization and DC semi-mature differentiation. However, there are no specific studies on the mechanisms (receptors involved, pathway activation/inactivation, metabolic activity, protein expression, and gene activation/inactivation) by which these natural compounds could exert such immunomodulatory effects on the APC.

## 3. Materials and Methods

### 3.1. Obtaining Plant Material

The cultivation and collection of the plant material of *Bidens pilosa* L. was carried out in the Jorge Piñeros Corpas Medicinal Garden of the Juan N. Corpas University Foundation, in Bogotá D.C, Colombia. The taxonomic identification of the plant was carried out in the national herbarium of the Faculty of Sciences of the National University of Colombia (registration number, 609178). The use of this plant for research purposes was approved by the Ministry of Environment and Development sustainable in Colombia, under the resolution of access to genetic resources number 315 of 2021, framework Permit to Collect Specimens of Wild Species of Biological Diversity for Non-Commercial Scientific Research Purposes Resolution No. 00596 of 2018. The aerial part of the plant (leaves, stems, and flowers) were dried at 24–26 °C under aseptic conditions, then pulverized in a mill, packed, and stored in the dark at an average temperature of 19 °C.

### 3.2. Obtaining Extracts and Fractions

The pulverized dry aerial parts from *B. pilosa* sample (500 g) were mixed with petroleum ether (4 L) for fifteen days to separate the petroleum-ether-soluble and petroleum-ether-insoluble phases. The benzine-insoluble phase (more polar) was then subjected to ethanolic extraction (4 L) using a Soxhlet unit. Then, this ethanolic extract was fractionated using continuous liquid–liquid fractionation with solvents of increasing polarity. As a result, *B. pilosa* fractions in n-hexane, chloroform, ethyl acetate, and the hydroalcoholic residue (ethanol-water 1:1) were produced. All extracts and fractions were dissolved in dimethyl sulfoxide (DMSO).

### 3.3. Screening of the Immunomodulatory Activity of Bidens pilosa L. in Human PBMCs

#### 3.3.1. Isolation and Culture of Human Peripheral Blood Mononuclear Cells

Human PBMCs were isolated from buffy coats of blood units donated to the blood bank of the District Institute of Science, Biotechnology, and Innovation in Health, Bogotá, Colombia. Buffy coats were diluted in phosphate-buffered saline 1X (PBS 1X) and centrifuged in Ficoll-Paque™ Premium (Cytiva, Uppsala, Sweden) density gradients (1.084 g/mL) at 800 gravities for 40 min at 20 °C to isolate PBMCs. Cells were washed and resuspended in Dulbecco’s modified Eagle’s medium (DMEM) (Lonza, Visp, Switzerland) supplemented with 10% fetal bovine serum (FBS) (Eurobio, Les Ulis, France), 100 U/mL penicillin, 100 µg/mL streptomycin (Corning Mediatech, Corning, NY, USA, EE. UU.), and 1 mM sodium pyruvate (Sigma Aldrich, St. Louis, MO, USA, EE. UU.). Cells were counted with trypan blue in the Neubauer chamber and the cells with a greater viability (90%) were used for cytotoxicity and cell proliferation assays.

#### 3.3.2. Cytotoxicity Assays on PBMCs

PBMCs were cultured in 96-well plates (100,000 cells per well) and treated with different doses (1.56 µg/mL to 200 µg/mL) of extracts and fractions of *B. pilosa* at 37 °C, 5% CO_2_, for 12, 24, and 48 h. PBMCs exposed only to DMSO (vehicle control), untreated cells, or PBMCs stimulated with PHA at 0.04 μg/mL (PHA, MICROGEN LABG&M, Bogotá, Colombia) or blank reagent (well without cells), or cells treated with CUR (Sigma Aldrich, EE. UU.) or Δ9-THC (Restek, Bellefonte, PA, USA, EE. UU.) at different doses (0.015, 0.0045, and 0.135 and 3, 9, and 27 µg/mL, respectively) were run in parallel as controls. The latter were used as immunomodulatory (antiproliferative and anti-inflammatory) controls in the assays. Two independent experiments were carried out, and each concentration was tested in triplicate.

After incubation, culture media were replaced with 100 μL medium without phenol red. Then, 50 μL of 3-(4,5-dimethylthiazol-2-yl)-2,5-diphenyltetrazole bromide (MTT; Sigma, St. Louis, MO, USA, EE. UU.) dissolved in PBS 1X was added, and cells were incubated at 37 °C, 5% CO_2_ for 4 h. The resulting formazan crystals were dissolved in DMSO before reading absorbance (Abs) at 550 nanometers (nm) on an AccuReader microplate reader, model M965 (Metertech, Taiwan). The percentages of viable cells were calculated as
(Abs of the cells that receive the treatments − Abs reagent blank) × 100)/(Abs control with DMSO − Abs reagent blank)

With these values, the inhibitory concentration 50 (IC_50_) for each of the plant extracts and fractions was calculated.

Although there is no agreement on IC_50_ values specifically indicating the exact cytotoxic figure [116], different authors consider that an IC_50_ < 100 μg/mL could be considered a cytotoxic concentration [117,118]. A two-way ANOVA test was used to compare the mean values for each treatment and the vehicle control, using the GraphPad Prism software version 9.0 (GraphPad Prism Inc., San Diego, CA, USA).

#### 3.3.3. Cell Proliferation Assays on PBMCs

The synergistic or antagonistic effect of *B. pilosa* on cell proliferation was evaluated on human PBMC cultures stimulated with PHA at 0.04 μg/mL (PHA, LABG&M, Colombia), according to the standardization of this methodology in our laboratory, and treated with the plant extracts or fractions. Two types of experiments were performed: In the first, PBMCs were treated simultaneously with PHA and extracts or fractions of *B. pilosa* at different doses (1.56 to 200 µg/mL) for 12, 24, and 48 h. In the second type of experiment, the PBMCs were stimulated with PHA (0.04 μg/mL) and allowed to proliferate for 24 h before the addition of plant extracts or fractions, and then, they were incubated with these treatments for 12, 24, and 48 h. For each type of experiment, proliferation (PHA), vehicle (PHA and DMSO), cells treated with CUR or THC at different doses (0.015, 0.0045 and 0.135 and 3, 9, and 27 µg/mL, respectively), and reagent blank control cultures were run in parallel. Two independent experiments were carried out, and each concentration was tested in triplicate.

Cell proliferation was assessed via the MTT colorimetric assay as described above. The proliferation index (PI) was calculated using the following equation:PI = (Abs of treated cells − Abs of the reagent blank)/Abs of the vehicle control.

### 3.4. Measurement of Cytokines in Culture Supernatants

The levels (pg/mL) of IL-1β, IL-6, IL-8, IL-10, IL-12p70, and TNF released in the supernatants of cultures were quantified using the commercial BD Cytometric Bead Array (CBA) Human Inflammatory Cytokine kit (Becton, Dickinson and Company BD Life Sciences-Biosciences, Franklin Lakes, NJ, USA, EE. UU.) in FACSAria II (Becton Dickinson, Franklin Lakes, NJ, USA, EE. UU.) citometer and FCAP array software v 3.0 (BD, Bioscience, Franklin Lakes, NJ, USA, EE. UU.) for analysis. A one-way ANOVA test was used to evaluate differences between levels of cytokines using GraphPad Prism software version 9.0.

### 3.5. Analysis of the Immunomodulatory Activity of B. pilosa L. on Human MØs and CDs

#### 3.5.1. Culture and Differentiation of Human Monocyte-Derived MØs

Human CD14^+^ monocytes were isolated from PBMCs using the Human CD14 Beads, QuadroMACS Starting Kit (LS) (No. 130-091-051) (Miltenyi Biotec GmbH, Bergisch, Gladbach, Germany) according to the manufacturer’s instructions. The isolated monocytes were then cultured to be differentiated into MØs or DCs.Briefly, monocytes (700,000 cells per well) were cultured in 12-well plates with RPMI medium and 20 ng/mL of granulocyte-monocyte colony-stimulating factor (GM-CSF, R&D system, Minneapolis, MN, USA, EE. UU.) that favors M1 polarization (M1-preconditioned MØs), or 20 ng/mL of macrophage colony-stimulating factor (M-CSF, R&D system, EE. UU.) that favors M2 polarization (M2-preconditioned MØs) at 37 °C, 5% CO_2_ for six days, replacing the culture media on the third day [119,120].

The expression of common macrophage markers was evaluated on the sixth day with antibodies: CD16 APC/CY7 (Biolegend, San Diego, CA, USA, EE. UU.), CD163 PERCP (Biolegend, EE. UU.), and CD86 PE (BD Pharmingen™, Franklin Lakes, NJ, USA, EE. UU.), using FACSAria II cytometer (Becton Dickinson, EE. UU.).

At the sixth day, M1- and M2-preconditioned MØs were then exposed to *B. pilosa* extracts or selected fractions (40 µg/mL according to the activity observed in PBMCs) in duplicate at 37 °C, 5% CO_2_ for 48 h.

M1- and M2-preconditioned MØ (non-polarized or untreated), polarized M1 (addition of interferon gamma (IFN-γ at 40 ng/mL) to the M1-preconditioned MØ culture on the sixth day) (R&D system, EE. UU.), polarized M2 (addition of IL-4 at 40 ng/mL to the M2-preconditioned MØ culture on the sixth day) (R&D system, EE. UU.), vehicle (addition of DMSO on the sixth day), and CUR at 15 ng/mL controls were run in parallel. Two independent experiments were carried out in duplicate.

The ability of the petroleum ether extract of *B. pilosa* to modulate or reprogram M1 polarizing toward M2 was also evaluated. To this purpose, M1-preconditioned MØs were treated with IFN-γ at the sixth day for 48 h and were then treated with the petroleum ether extract of *B. pilosa* at 40 µg/mL or DMSO (vehicle control) or CUR at 15 ng/mL in duplicate for an additional 48 h at 37 °C, 5% CO_2_.

The concentration tested in duplicate was selected according to the activity observed in M1- or M2-preconditioned MØs. Staining conditions of these cells were created and analysis was carried out in the same way as previously described.

#### 3.5.2. Culture and Differentiation of Human Monocyte-Derived DCs

CD14^+^ human monocytes (600,000 cells per well) were cultured in 12-well plates with RPMI medium (Lonza, Switzerland) and 100 ng/mL GM-CSF and 20 ng/mL IL-4 (R&D system, EE. UU.) at 37 °C, 5% CO_2_ for five days, replacing the culture media on the third day. 

The immature phenotype of these DCs (CD206^high^, CD209^high^, CD86^low^, and HLA-DR^low^) was determined via flow cytometry using CD206 PE/CY7 (Biolegend, EE. UU.), CD209 PERCP/CY5 (BD Pharmingen™, EE. UU.), and HLA-DR FITC (Biolegend, EE. UU.) antibodies using a FACSAria II cytometer (Becton Dickinson, EE. UU.).

Immature DCs were then treated with the extracts and fractions of *B. pilosa,* selected in duplicate, at 37 °C, 5% CO_2_ for 48 h. Immature DC (untreated), mature DC (addition of lipopolysaccharide [LPS] 1 μg/mL on the fifth day) (Sigma-Aldrich, EE. UU.), immature cells treated with CUR at 15 ng/mL, or vehicle (addition of DMSO on the fifth day) controls were run in parallel in two independent experiments.

The capability of the petroleum ether extract of *B. pilosa* to modulate the maturation process of DCs in an inflammatory context was also evaluated. To this end, immature DCs (derived from monocytes cultured with GM-CSF and IL-4 for five days) were treated on the fifth day with LPS (1 μg/mL) (maturating stimulus), LPS and either petroleum ether extract of *B. pilosa* or DMSO (vehicle control) at the same time, or CUR at 15 ng/mL in duplicate and further incubated at 37 °C, 5% CO_2_ for 48 h.

#### 3.5.3. Characterization of the Phenotype of MØs and DCs

After incubation, MØs and DCs were phenotypically characterized by assessing cytokine synthesis and surface marker expressions. For this purpose, culture supernatants were collected for cytokine quantitation using the CBA Human Inflammatory Cytokine kit (Becton, Dickinson and Company BD Life Sciences-Biosciences, EE. UU.) and analyzed via flow cytometry using FACSAria II (Becton Dickinson, EE. UU.) and FCAP array software v 3.0 (BD, Bioscience, EE. UU.). A one-way ANOVA test was used to evaluate differences between levels of cytokines of treated cells and DMSO-treated cells using GraphPad Prism software version 9.0.

Additionally, MØs and DCs were re-collected and stained with fluorochrome-conjugated antibodies and analyzed via flow cytometry. First, Fc receptors were blocked using Human TruStain FcX™ (Biolegend, EE. UU.) according to the manufacturer’s instructions. Then, MØs were stained with anti-CD16 Alexa Fluor 700 (BD Pharmingen™, EE. UU.), anti-CD68 FITC (Biolegend, EE. UU.), anti-CD86 PE (BD Pharmingen™, EE. UU.), anti-iNOS Alexa fluor 594 (Novus Biologicals™, Centennial, CO, USA, EE. UU.), anti-CD206 PE/CY7 (Biolegend, EE. UU.), anti-CD209 PERCP (BD Pharmingen™, EE. UU.), and anti-CD163 PE (Biolegend, EE. UU.) antibodies to assess the M1 profile (CD16^High^, CD68^High^, CD86^High^, iNOS^High^) and M2 profile (CD16^High^, CD206^High^, CD209^High^, CD163^High^).

Similarly, DCs were stained with anti-CD86 PE (BD Pharmingen™, EE. UU.), anti-HLA-DR FITC (BD Pharmingen™, EE. UU.), anti-CD83 Alexa Fluor 700 (Invitrogen, Waltham, MA, USA, EE. UU.), anti-CD206 PE/CY7, and anti-CD209 Percp-Cy5 (BD Pharmingen™, EE. UU.) antibodies to assess their immature (CD206^High^, CD209^High^, CD86^Low^, CD83^Low^, and HLA-DR^Low^), semi-mature (CD206^high^, CD209^high^, CD86^high^, CD83^high^, and HLA-DR^high^), and mature (CD206^low^, CD209^low^, CD86^high^, CD83^high^, and HLA-DR^high^) profile.

Stained cells were acquired on a FACSAria II flow cytometer (Becton Dickinson), and data were analyzed using the FlowJo™ Software version 10.0 (FlowJo, Ashland, OR, USA: Becton, Dikinson and Company, Franklin Lakes, NJ, USA). Results were expressed as mean ± SD. Mean values of cultures exposed to *B. pilosa* and of vehicle control cultures were compared by a two-way ANOVA test using the GraphPad Prism software, v. 9.0.

### 3.6. Chemical Characterization through Gas Chromatography Coupled with Mass Spectrometry (GC/MS)

Chromatographic profiling and mass spectra analyses of *B. pilosa* of petroleum ether extract were carried out using the Shimadzu single quadrupole GCMS-QP2020 NX gas chromatograph–mass spectrometer (Shimadzu Scientific Instruments, Kyoto, Japan, MD, EE. UU.), equipped with an HP5-MS capillary column (30 m in length × 0.25 mm in diameter × 0.25 µm in thickness) (Agilent J&W Scientific, Palo Alto, CA, USA, EE. UU.). Helium gas was used as the carrier gas at a constant 1 mL/min flow rate. For GC-MS spectral detection, the spectrometer was set up in positive ionization mode with electron impact (EI) at 70 eV, in full scan mode from 100 to 800 *m*/*z*. The injection quantity of 1 μL was used (10:1 split ratio), and the injector temperature was 280 °C.

The program used was set initially at 100 °C for 3 min, gradually raised at 15 °C/min up to 220 °C, which was maintained for 3 min, and then raised at 8 °C/min up to 300 °C for 10 min. The phytochemical constituents present in the extract were tentatively identified by their retention time (RT) (min), peak area, and peak height, and by comparing mass spectral patterns with the spectral data of authentic compounds stored in the National Institute of Standards and Technology (NIST) 08 mass spectral library with a similarity percentage greater than 80%.

## 4. Conclusions

Extracts and fractions of different polarities obtained from *B. pilosa* were found to be slightly cytotoxic or non-cytotoxic to primary human cells (PBMCs and APCs). This characteristic, together with their anti-inflammatory properties, makes them a promising source of compounds to be considered in developing new immunomodulatory therapies to regulate the response of immune system cells under pathological conditions.

The low–medium polarity products of *B. pilosa* (petroleum ether extract and ethyl acetate fraction) showed better antiproliferative activity on human PBMCs when added concomitantly with PHA than when added to already proliferating PBMCs.

The most hydrophobic extract (petroleum ether) of *B. pilosa* exerted the most significant antiproliferative effect on PHA-stimulated PBMCs and was the most efficient at promiting MØ polarization into the M2 phenotype and at driving DCs to acquire a semi-mature status. These in vitro results indicate that the petroleum ether extract of *B. pilosa*, enriched in immunomodulatory compounds such as terpenes and fatty acids, could down-regulate the immune response by acting on different cells involved in innate and adaptive immunity.

These results were supported by the finding that this extract induced M1-preconditioned MØs to express an M1/M2 hybrid phenotype and semi-mature DCs to remain as such despite being exposed to LPS.

All the findings described above allow us to consider future research focused on the therapeutic application of the anti-inflammatory properties of *B. pilosa*-derived products in inflammatory settings.

## Figures and Tables

**Figure 1 molecules-28-07094-f001:**
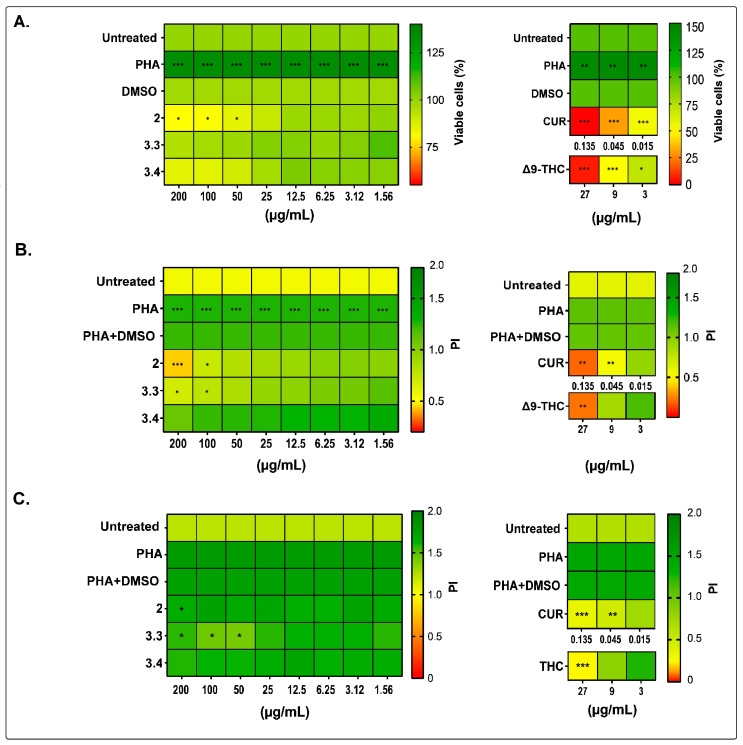
Effects of *B. pilosa* L. on human PBMCs. (**A**). Heat maps of percentage (%) of viable PBMCs treated at different concentrations of petroleum ether extract (2), ethyl acetate fraction (3.3), or hydroalcoholic fraction (3.4) of *B. pilosa* and cultured at 37 °C, 5% CO_2_ for 48 h. Control cultures were prepared with untreated PBMCs, PBMCs treated with curcumin (CUR) or delta-9-tetrahydrocannabinol (Δ9-THC), or DMSO (vehicle control). The percentage of viable cells relative to the vehicle control was determined via the colorimetric MTT assay. (**B**). Heat maps of proliferation index (PI) of PBMCs were stimulated with PHA (0.04 μg/mL) and simultaneously exposed to different concentrations of CUR, Δ9-THC or petroleum ether extract (2), or ethyl acetate fraction (3) or hydroalcoholic fraction (4) of *B. pilosa* and cultured at 37 °C, 5% CO_2_ for 48 h. (**C**). Heat maps of PI of PBMCs that were stimulated with PHA (0.04 μg/mL) at 37 °C, 5% CO_2_ for 24 h, and then exposed to the extracts of *B. pilosa* or controls, as described in (**B**). Then, the PI was determined via the MTT colorimetric assay, relative to the respective vehicle control. Statistical significance for all experiments was obtained using two-way ANOVA and Dunnet’s post hoc multiple comparison test (* *p* < 0.05, ** *p* < 0.01, *** *p* < 0.001).

**Figure 2 molecules-28-07094-f002:**
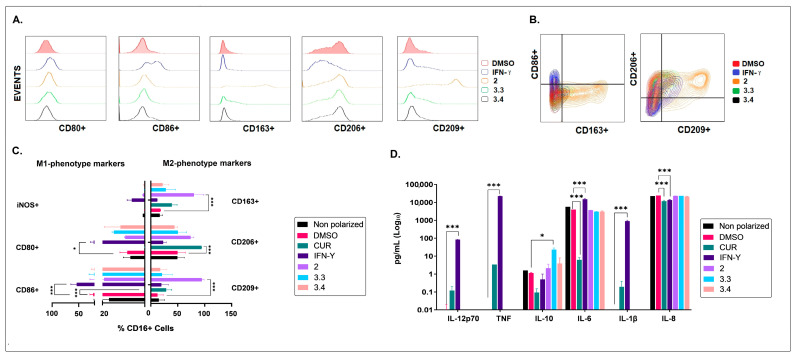
Effects of the petroleum ether extract and the ethyl acetate and hydroalcoholic fractions of *B. pilosa* on the polarization of M1-preconditioned MØs. Human M1-preconditioned MØs (derived from monocytes cultured with 20 ng/mL of GM-CSF for six days) were exposed to petroleum ether extract (2), the ethyl acetate fraction (3.3), or the hydroalcoholic fraction (3.4) of *B. pilosa* and cultured at 37 °C, 5% CO_2_ for 48 h. M1-preconditioned MØ (non-polarized or untreated), polarized M1 (addition of IFN-γ at 40 ng/mL on the sixth day), M1-preconditioned treated with curcumin (CUR) at 15 ng/mL (immunomodulator control), and vehicle (addition of DMSO on the sixth day) MØ culture controls were run in parallel. Macrophage phenotype was assessed by staining with fluorescent antibodies and flow cytometry. Culture supernatants were collected for cytokine quantification using CBA kit and flow cytometry. (**A**). Representative histograms comparing MØ phenotypic markers. (**B**). Representative contour plots showing co-expression of MØ phenotypic markers. (**C**). Column graph shows the percentages (mean ± SD) of MØ (CD16^+^ cells) expressing M1 or M2 phenotypic markers. (**D**). Column graph shows the cytokine levels in culture supernatants (mean ± SD). * *p* < 0.05, *** *p* < 0.001. One-way or two-way ANOVA tests and Dunnet’s post hoc multiple comparison test. *n* = 2.

**Figure 3 molecules-28-07094-f003:**
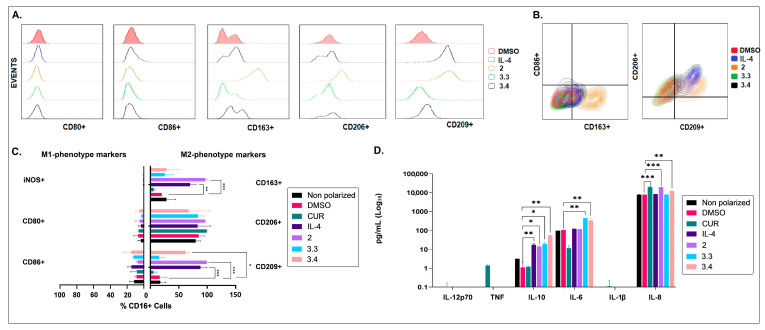
Effects of the petroleum ether extract and the ethyl acetate and hydroalcoholic fractions of *B. pilosa* on the polarization of M2-preconditioned MØs. Human M2-preconditioned MØs (derived from monocytes cultured with 20 ng/mL of M-CSF for six days) were exposed to petroleum ether extract (2), the ethyl acetate fraction (3.3), or the hydroalcoholic fraction (3.4) of *B. pilosa* and cultured at 37 °C, 5% CO_2_ for 48 h. M2-preconditioned (non-polarized or untreated), polarized M2 (addition of IL-4 at 40 ng/mL on the sixth day), M2 treated with curcumin (CUR) at 15 ng/mL (immunomodulator control), and vehicle (addition of DMSO on the sixth day) MØ culture controls were run in parallel. Macrophage phenotype was assessed by staining with fluorescent antibodies and using flow cytometry. Culture supernatants were collected for cytokine quantification using CBA kit and flow cytometry. (**A**). Representative histograms comparing MØ phenotypic markers. (**B**). Representative contour plots showing co-expression of MØ phenotypic markers. (**C**). Column graph shows the percentages (mean ± SD) of MØ (CD16^+^ cells) expressing M1 or M2 phenotypic markers. (**D**). Column graph shows the cytokine levels in culture supernatants (mean ± SD). * *p* < 0.05, ** *p* < 0.01, *** *p* < 0.001. One-way or two-way ANOVA tests and Dunnet’s post hoc multiple comparison test.

**Figure 4 molecules-28-07094-f004:**
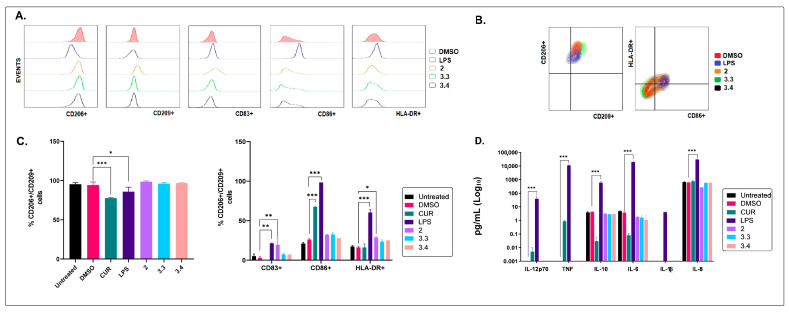
Effects of the petroleum ether extract and the ethyl acetate and hydroalcoholic fractions of *B. pilosa* on the maturation status of human DCs. Human immature DCs were exposed to petroleum ether extract (2), the ethyl acetate fraction (3.3), or the hydroalcoholic fraction (3.4) of *B. pilosa* at 37 °C, 5% CO_2_ for 48 h. Immature (untreated), mature (addition of LPS at 1 μg/mL on the fifth day), treated with curcumin (CUR) at 15 ng/mL (immunomodulator control), and vehicle (addition of DMSO on the fifth day) DC control cultures were run in parallel. Dendritic cell phenotype was assessed by staining with fluorescent antibodies and using flow cytometry. Culture supernatants were collected for cytokine quantification using CBA and flow cytometry. (**A**). Representative histograms comparing DC phenotypic markers. (**B**). Representative contour plots showing co-expression of DC phenotypic markers. (**C**). Column graphs show the percentages (mean ± SD) of total CD206^+^CD209^+^ DCs and CD206^+^CD209^+^ subsets in the CD83^+^, CD86^+^, or HLA-DR^+^ DCs. (**D**). Column graphs show cytokine levels (mean ± SD) in culture supernatants. * *p* < 0.05, ** *p* < 0.01, *** *p* < 0.001. One-way or two-way ANOVA tests and Dunnet’s post hoc multiple comparison test.

**Figure 5 molecules-28-07094-f005:**
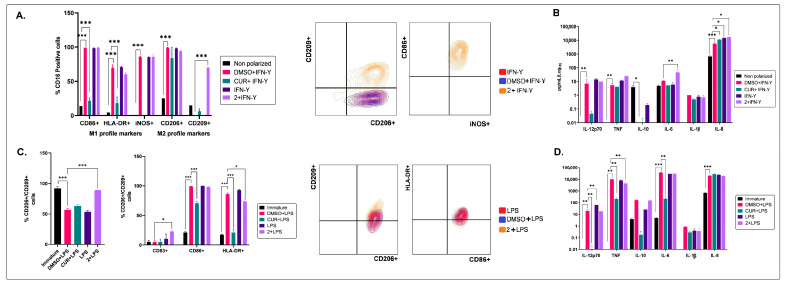
Effects of the petroleum ether extract of *B. pilosa* on M1 polarization of macrophages and maturation of LPS-stimulated dendritic cells. (**A**). Human M1-preconditioned MØs were exposed to IFN-γ at 40 ng/mL (M1-polarizing stimulus) for 48 h and then treated with curcumin at 15 ng/mL (CUR) (immunomodulator control), petroleum ether extract (2) of *B. pilosa,* or DMSO (vehicle control) for an additional 48 h, incubated at 37 °C, 5% CO_2_. Column graphs show the percentages (mean ± SD) of MØs (CD16^+^ cells) expressing M1 or M2 phenotypic markers. Also, representative contour plots of MØs co-expressing M1 or M2 markers are shown. (**B**). Column graphs show cytokine levels (mean ± SD) in MØ culture supernatants. (**C**). Human immature DCs were exposed to LPS at 1 μg/mL (maturating stimulus) and the CUR at 15 ng/mL or petroleum ether extract (2) of *B. pilosa* at the same time, or DMSO (vehicle control) on fifth day and further incubated at 37 °C, 5% CO_2_ for 48 h. DCs were evaluated by staining with fluorescent antibodies and using flow cytometry. Column graphs show the percentages (mean ± SD) of total CD206^+^CD209^+^ DCs and CD206^+^CD209^+^ subsets in the CD83^+^, CD86^+^, or HLA-DR^+^ DCs. Also, representative contour plots of DCs co-expressing phenotypic markers are shown. (**D**). Column graphs show cytokine levels (mean ± SD) in DC culture supernatants (* *p* < 0.05, ** *p* < 0.01, *** *p* < 0.001). One-way or two-way ANOVA tests and Dunnet’s post hoc multiple comparison test.

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
