# Peer review of "A Bidens pilosa L. Non-Polar Extract Modulates the Polarization of Human Macrophages and Dendritic Cells into an Anti-Inflammatory Phenotype"

_molecules, 2023, doi:10.3390/molecules28207094_

Round 1

Reviewer 1 Report

In this study, the authors covered the gap in the ability of well-known immunomodulatory world-traditional Bidens pilosa L. to modulate antigen-presenting cells; macrophages, and dendritic cells.  The study showed that petroleum ether extract has the most significant anti-proliferative effect on  PHA-stimulated PBMCs, promotes macrophage polarization towards M2, and drives dendritic cells to semi-mature status. This extract might have therapeutic potential in the treatment of inflammatory and autoimmune diseases. 

Minor comments: 

1- In the introduction: please add the reference for the second paragraph.

2- In figure 1, please fix IP to be PI (proliferation index)

Author Response

Consulte el archivo adjunto.

Reviewer 2 Report

This article in title” A Bidens pilosa L. non-polar extract modulates polarization of human macrophages and dendritic cells to an anti-inflammatory phenotype” demonstrates a viewpoint for the immunomodulatory and anti-inflammatory roles of Bidens pilosa L. non-polar extract in human macrophages. The article is interesting. However, there are four concerns needed to be addressed further. 

1. The experimental designs lack a positive control. Authors should select a known drug which expresses an immunomodulatory activity to compare with the B. pilosa extracts.

2. Aqueous infusion and methanolic extracts of Bidens pilosa has been studied the immunomodulatory activity in whole blood cells and lymophocytes (J Ethnopharmacol 2004 Aug;93(2-3):319-23. and Immunopharmacology 1999 Jun;43(1):31-7.). Authors should discuss these differences compare with these studies.

3. The authors used colorimetry to display cell viability. It cannot see precise quantitative data. Authors should add a dotted line graph to represent cell viability.

4. In Figure 1B and 1C, some bar graphs lack bar of untreated groups. Authors should recheck.

Round 2

Reviewer 2 Report

The author has responded to the review comments in detail. I recommend accepting this article for publication.